# Control of Maternal-to-Zygotic Transition in Human Embryos and Other Animal Species (Especially Mouse): Similarities and Differences

**DOI:** 10.3390/ijms23158562

**Published:** 2022-08-02

**Authors:** Jan Tesarik

**Affiliations:** MARGen Clinic, 18006 Granada, Spain; jtesarik@clinicamargen.com; Tel.: +34-606-376-992

**Keywords:** maternal-to-zygotic transition, zygotic gene activation, maternal mRNA decay, M-decay, Z-decay, DNA transcription, RNA translation, embryo developmental competence, human assisted reproduction

## Abstract

Maternal-to-zygotic transition (MZT) of the control of early post-fertilization development is a key-event conditioning the fate of the future embryo, fetus and newborn. Because of the relative paucity of data concerning human embryos, due to ethical concerns and the poor availability of human embryos donated for research, most data have to be derived from animal models, among which those obtained using mouse embryos are most prevalent. However, data obtained by studies performed in non-mammalian specie can also provide useful information. For this reason, this review focuses on similarities and differences of MZT control mechanisms in humans and other species, with particular attention to the mouse. A number of molecular pathways controlling MZT in mice and humans are compared, pointing out those that could be at the origin of further focused experimental studies and the development of new diagnostic tools based on the translational medicine principles. Data concerning possible candidate molecules to be included in these studies are identified.

## 1. Introduction

Maternal-to-zygotic transition (MZT) refers to an initial step in the early embryonic development, consisting of two strictly coordinated phenomena, i.e., zygotic genome activation (ZGA) and the degradation of maternal gene transcripts. MZT is observed in all animal species investigated so far [1] including mammals [2]; it is essential for embryonic development because it coordinates cell division and allocation to the first two cell lineages, i.e., the inner cell mass (ICM) and trophectoderm, at the blastocyst stage [3]. Even though, so far as human embryos are concerned, there are still a lot of open questions, there is no doubt that failure of MZT is involved in preimplantation embryo demise and the resulting failure of assisted reproduction techniques (ART) [4]. Knowledge of the molecular mechanisms regulating MZT in humans is thus important for the development of diagnostic methods with which the quality of each individual embryo can be tested and the adequacy of particular ovarian stimulation protocols in each individual woman can be evaluated, enabling an individualized approach [5].

Because of the limited availability of human embryos to be used for scientific investigation, researchers have tried to extrapolate data obtained in various animal species to humans. However, it soon became evident that this approach was complicated, due to many significant interspecies differences in the mechanism and timing of MZT. Most studies were performed in different rodent species, mainly the mouse. However, the timing of human MZT is similar to that of some other mammals, including cow, sheep, rabbit, and macaque [6,7,8,9], suggesting that data obtained in these species may be more representative of humans [3].

Nevertheless, the mouse is the most available experimental animal model for the study of early embryonic development, and as such, data obtained with this model are important as a source of basic knowledge to be tested and appraised in a human clinical context, albeit while taking into account the interspecies differences [4]. Last but not least, a lot of studies have been published using non-mammalian animal species. Surprisingly, though phylogenetically more distant from humans, some molecular mechanisms of early embryonic development, including those related to MZT, are quite similar to those in humans; therefore, data obtained in those species should also be taken into account.

This article outlines the principal findings obtained in human embryos and confronts them with those coming from animal studies in order to suggest directions for future research and the development of new diagnostic and therapeutic methods to be used in the management of human infertility.

## 2. Similarities and Particularities of MZT in Human Preimplantation Embryos as Related to Other Animal Species

There are two classes of vertebrates: anamniotes and amniotes. The former lay oocytes externally in water and include bony fish and amphibians, whereas the latter lay fertilized oocytes on land or retain them in the mother in a protective membrane impermeable to water; these include mammals, birds and reptiles [3]. Anamniotic embryos contain all they need for the early development and typically show a period of rapid cell division following fertilization, whereas amniotic embryos develop more slowly [3]. In spite of this difference, some conserved principles of the vertebrate MZT appear to exist in both anamniotes and amniotes. Anamniotic embryos are also called fast-developing embryos, as opposed to the slow-developing amniotic embryos. In fast-developing embryos, early rapid synchronous cell divisions do not require embryonic gene transcription until gastrulation, and their cell cycles often lack G1 and G2 phases, alternating directly between the S and M phases [3].

A comparison of molecular events taking place during preimplnatation embryo development in human and mouse shows some remarkable differences (Figure 1), but also many similarities (Figure 2) which can serve to design future molecular studies aimed at developing new diagnostic and therapeutic strategies in the management of human infertility.

In terms of generating a complete chain of gene transcription and expression, the beginning of ZGA in mammalian embryos occurs at the two-cell stage in mice [3], at the four-cell stage in pigs [10], at the eight-cell stage in sheep [6], cows [7] and rabbits [8], and between the four- cell and eight-cell stage in macaques [9] and humans [11,12,13]. In this sense, the early ZGA occurring in mice is rather exceptional, and its even later onset was also described in a variety of vertebrate anamniotic species (reviewed in [3]). However, the first detectable signs of gene transcription occur as early as the one-cell stage in many mammalian species, including mouse [14] and human [15], although most of these early transcripts are not translated into proteins, and their main function is to prepare embryos for the subsequent ZGA. In addition to participating in the nucleologenesis required for the production of the ribosomes needed during the forthcoming major wave of protein synthesis [15], this early wave of gene activity is also the source of other functional RNA coding for transcriptional factors participating both in the regulation of ZGA and maternal RNA degradation [16].

Before the onset of MZT, the early developmental processes in preimplantation embryos are entirely under the control of maternally inherited factors present in the mature oocyte. In fact, maternal RNAs and proteins exclusively guide the development while the zygotic genome remains quiescent [17]. Embryos of mammalian species that display early MZT onset, such as the mouse, are thus likely to be less dependent on maternal RNAs and proteins stored in the unfertilized oocyte as compared to species with a relatively late MZT onset, such as the human. This can explain why developmental competence has not yet been achieved with the use of in vitro culture systems to support preantral follicle development up to maturation in human and domestic animals [18]. The adequate progression of early post-fertilization developmental events occurring before the beginning of MZT is dependent on timely translation regulation of maternal mRNAs in time and space in all species studied so far [19]. The mechanisms involved in these regulatory processes, revealed in different species, involve mRNA association with, and its subsequent orchestrated release from, RNA-binding proteins [20], the spatial localization of the RNA-protein (RNP) complexes in the cytoplasm [21], and the timely activation of specific mRNA translation by modulation of cytoplasmic polyadenylation [22,23].

Both ZGA and maternal RNA clearance, the two basic events occurring during MZT, are highly complex processes that have to occur in a strictly orchestrated way in order to ensure normal embryonic development [24]. In *Drosophila*, maternal RNA clearance was shown to be a necessary pre-requisite for ZGA and MZT [25]. Studies on Drosophila [26] and mouse [27] embryos have shown that the elimination of maternal transcripts is accomplished by two sequential pathways. The first pathway, termed maternal decay (M-decay), is entirely mediated by maternal factors accumulated in the mature oocytes, whereas the second pathway, termed zygotic decay (Z-decay), depends on de novo zygotic transcription products appearing after fertilization. Key factors regulating M-decay and Z-decay pathways appear to have similar expression patterns in mouse and human MZT, involving YAP1-TEAD4 transcription activators, TUT4/7-mediated mRNA 3′-oligouridylation, and BTG4/CCR4-NOT-induced mRNA deadenylation [24]. The observation that homozygous mutations in human BTG4 cause zygotic cleavage arrest and female infertility [28] supports the hypothesis of functional equivalence of the pathways controlling M- and Z-decay in mice and humans. Both M- and Z-decay pathway activities appear to contribute to the developmental potential of human preimplantation embryos [24].

Despite extensive research on murine maternal and zygotic transcriptomes in recent years, it is still not clear to what extent the transcriptional pathways of mouse and human embryos are similar, and many questions regarding key MZT events in humans remain unanswered [29]. For instance, the proportion of human maternal transcripts with ZGA-dependent clearance remains undetermined [24]. In addition to similarities between the MZT regulating pathways in mice and humans, some differences are also likely to exist. The discovery of hominoid specific transposable elements and KZFPs that control human embryonic genome activation [30] points in this direction. Other differences between the early embryos of mice and humans concern the dynamics of the DNA methylation pattern, a crucial element in the epigenetic regulation of mammalian embryonic development. In fact, the major wave of genome-wide demethylation in human embryos is complete at the two-cell stage. Contrary to observations in mice, the demethylation of the paternal genome is much faster than that of the maternal genome, so the genome-wide methylation level in male pronuclei is already lower than that in female pronuclei [31].

Moreover, comparisons of human and mouse transcriptomes during MZT revealed that only half of the transcriptomes were overlapping, indicating that the homology between human and mouse maternal transcriptomes is low, and even fewer zygotically activated genes are shared by mouse two-cell embryos and human eight-cell embryos [24]. Also, some transcripts degraded by M-decay pathways in humans were found to be degraded by Z-decay pathways in mice, and vice versa, suggesting that subsets of human mRNA might be regulated differently from mouse during MZT [24]. In general, zygotic transcription plays a more important role in the elimination of human maternal transcripts, supposedly due to a longer time span from the onset of major ZGA to the completion of maternal mRNA decay in humans compared to mice [32].

As for the mechanisms regulating MZT timing, three models have been proposed. The first claims that MZT dynamics are triggered by the accumulation of maternally deposited activating transcription factors. The second model suggests that MZT dynamics are triggered by reaching a threshold ratio of a nuclear component to the cytoplasmic volume, often referred to as the “N/C-ratio”. The third model implies the de novo establishment of chromatin states which allow transcription to be critical for ZGA timing, because embryos must start from a state with no transcription. Importantly, these three timing mechanisms are not mutually exclusive and additional mechanisms may also exist [3].

Regarding mammalian embryos, maternally deposited activating transcription factors Nanog, SoxB1, and Oct4 (Pouf1) were suggested to activate transcription in preimplantation embryos of mice and human embryonic stem cells [33]. As for the “N/C-ratio” model, convincing data available for mammalian embryos are lacking. By contrast, the mechanisms implying the de novo establishment of chromatin states which are permissive for transcription are active in mammalian embryos, including humans (see below).

It is also important to note that the clearance timing differs among specific mRNA species, and there is a period of overlap during which both maternal and zygotic transcripts participate in the control of embryonic development. It was shown that in human embryos in which some blastomeres failed to undergo the major ZGA by the eight-cell stage, the development of intercellular junctions, required to seal the fluid-filled space in the future blastocyst, began in the same way in the cells that had accomplished ZGA and in those that had not [34]. However, this process was incomplete in ZGA-failed blastomeres. Thus, oocyte-coded messages are apparently involved in the control of the relatively late stages of human preimplantation development, including the differentiation of the first two embryonic cell lines, while the embryonic genome is required for the full achievement of this early differentiation event. In any case, a temporal overlap in the translation of maternal and zygotic transcripts is a common feature of embryos of different animal species [35].

The role of the double homeobox (DUX) family of transcription factors is also highly conserved among different mammalian species, including mouse and primates [36], where they act as pioneer actors of ZGA [37,38]. Mouse DUX and its human homolog DUX4 activate ZGA by acting as regulators of the non-coding genome [39].

Taken together, studies performed so far have revealed both similarities and differences in the control of MZT among mammalian species. Concerning possible future implications in human infertility management and research, the use of data obtained in mice, by far the most studied mammalian species, will certainly serve as the main basis.

## 3. Implications in Research and Clinical Issues Related to Human ART

More than 40 years after the first successful IVF attempt [40] and the subsequent introduction of many clinical and laboratory procedures extending IVF indication to a growing number of clinical conditions [41], the current success rate of all these ART variants remains relatively low [42]. The indiscriminate use of standardized clinical and laboratory protocols for all patients undergoing ART treatments is likely to be one of the major culprits. This is why there is an increasing need for personalizing diagnostic and therapeutic methods according to the condition of each infertile woman [5]. The concept of personalized medicine, also called precision medicine or molecular medicine, was applied with success in the management of gynecologic cancer and is currently gaining ground in the field of reproductive medicine, too [43].

The application of personalized medicine in ART depends, on the one hand, on the availability of high-quality molecular biology data applicable to the human species, and other other, on a thorough understanding of how to use these data in the clinical context. Unfortunately, most molecular biologists have only limited knowledge of clinical medicine, and similarly, most clinicians have not mastered basic concepts of molecular biology so as to be able to use data from molecular studies in their field. Hence, there is a growing need to apply the basic concept of translational medicine, “the bench-to-bedside” approach, to address the challenges currently faced in human infertility management.

This review is focused on the similarities and differences between the control of early embryonic developmental processes in mice—by far the most extensively studied mammalian species—and humans, as the potential target for clinical application of mouse-derived data. Despite some differences, the basic features of the early developmental control are shared between the two species, always taking into account the timing differences. Accordingly, mouse data obtained in two-cell embryos cannot be applied to human two-cell embryos but are relevant to human eight-cell embryos, where comparable changes in cell developmental control occur. With these differences in mind, many findings obtained in mice, such as the modulation of mRNA association with RNA-binding proteins, the main factors regulating M- and Z-decay pathways, and the key role of the DUX-family of transcription factors in the priming of embryonic genome activation, can be applied to humans (see Section 2 of this article for relevant references).

There are two ways in which data generated in mouse models can be used in human reproductive medicine. First, genomic, transcriptomic, proteomic and, to a limited degree, epigenomic analyses of human cells and fluids recovered during ART attempts can be performed to evaluate the adequacy of the ovarian stimulation protocols used, since it is known that different women need different ratios of gonadotropins and other hormones, such as growth hormone, for optimal ovarian stimulation [5]. This will help the clinician adapt the stimulation protocol to the specific needs of each patient in potential future attempts. Second, data which prove to be of importance for mouse embryonic development can be transposed to humans and serve as a basis for the elaboration of diagnostic tests with which to evaluate female fertility status.

There are also data obtained directly by studies on human embryos which can help understand some important developmental mechanisms determining the fate of each embryo. Early studies using ^3^H-uridine labeling and autoradiography revealed that many human embryos contain some blastomeres that had correctly activated gene transcription by the eight-cell stage, and other blastomeres that had failed to do so. This was especially the case of embryos containing some multinucleated blastomeres [44]. Compared with normal mononucleated blastomeres, the nuclei of the multinucleated blastomeres showed limited transcriptional activity, as reflected by ^3^H-uridine incorporation, but failed to show the same major increase in transcription as compared with mononucleated blastomeres [44]. In the original publication, blastomere multinucleation was interpreted as a sign of irreversible embryo damage. However, recent observations that the uterine transfer of embryos containing multinucleated blastomeres (1) can lead to ongoing pregnancies and births, though less frequently compared to the transfer of embryos with mononucleated blastomeres only, and (2) does not increase the occurrence of congenital anomalies and chromosomal defects [45], has led to another interpretation. It was suggested that blastomere multinucleation might represent a morphological correlation of the activation of a cell-cycle checkpoint which can convert a mosaic embryo to an euploid one [46].

Last but not least, it should be stressed that, in spite of its currently widespread use, preimplantation genetic testing for aneuploidies (PGTA) should be used with caution unless a high number of embryos are available for transfer. In fact, PGTA often gives false positive results that lead to unjustified wastage of potentially viable embryos, indisputably causing harm to the affected patients [47]. One reason for this is that the small number of cells recovered from embryonic trophectoderm for analysis do not necessarily reflect the chromosomal constitution of the inner cell mass which gives rise to the future embryo and fetus, especially in the case of mosaic embryos [48]. In fact, abnormal trophectoderm cells undergo clonal expansion, and the examination of such an abnormal clone can jeopardize the test interpretation [48]. Moreover, some embryos diagnosed as abnormal at day 5 of development can convert to normal ones in the following days, as demonstrated by the birth of normal offspring after the uterine transfer of embryos diagnosed as abnormal [49,50].

## 4. Conclusions

The mouse model is a rich source of molecular biology data relevant to preimplantation embryonic development, in general, and to MZT in particular. Despite some interspecies differences, many developmental mechanisms controlling MZT are similar in mice and humans. This makes it possible to use mouse data in future research in humans to obtain useful information related to the control of early human development and to develop diagnostic tools for evaluating the adequacy of ovarian stimulation protocols in each individual woman.

## Figures and Tables

**Figure 1 ijms-23-08562-f001:**
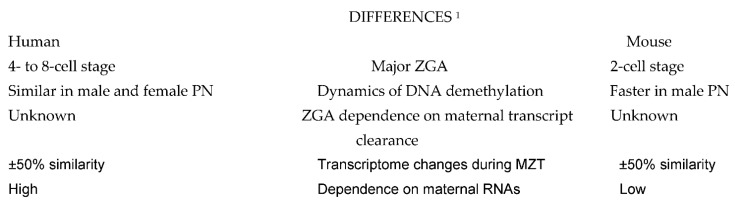
Differences between the basic molecular events taking place in human and mouse embryos. Explanations: ^1^ Comparisons are to be made taking into account the differences in ZGA timing. Abbreviations: ZGA: zygote gene activation; PN, pronuclei; MZT: maternal-to-zygotic transition of embryo developmental control.

**Figure 2 ijms-23-08562-f002:**
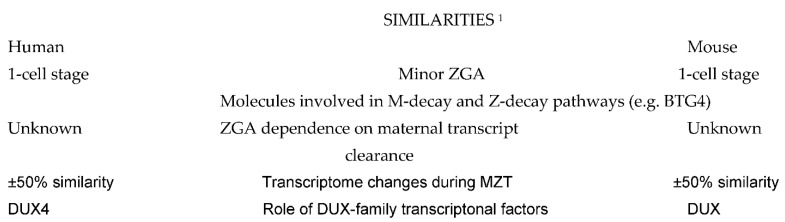
Similarities between the basic molecular events taking place in human and mouse embryos. Explanations: ^1^ Comparisons are to be made taking into account the differences in ZGA timing. Abbreviations: ZGA: zygote gene activation; MZT: maternal-to-zygotic transition of embryo developmental control; M-decay and Z-decay pathways: pathways of maternal transcript elimination controlled by maternal and zygotic transcripts, respectively. DUX: double homeobox.

## Data Availability

Not applicable.

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
