# Peer review of "Control of Maternal-to-Zygotic Transition in Human Embryos and Other Animal Species (Especially Mouse): Similarities and Differences"

_ijms, 2022, doi:10.3390/ijms23158562_

Round 1

Reviewer 1 Report

Maternal-to-zygotic transition (MZT) is one of the crucial processes in early development. The MZT disturbances, including in human, can cause serious disorders or even arrest of embryo development, which determines the great practical interest for MZT studies. The MS under review is devoted to the comparative analysis of MZT timing and regulatory mechanisms in early human development, with special emphasis on the comparison with other animals, primarily mice. The author discusses not only some differences between various MRT strategies, but also the current possibilities of using a mouse model to improve the efficiency of ART. The submitted MS, therefore, is of great interest to a wide range of specialists.

The MS is conceived and written clearly and precisely. No principal questions or comments arise after reading. Meanwhile, it would be helpful to add a schematic drawing or table that could summarize and highlight the similarities/differences between human and mouse MZT. Such a final summary would further enhance the value of the review for readers.

Author Response

Comments and Suggestions for Authors

Maternal-to-zygotic transition (MZT) is one of the crucial processes in early development. The MZT disturbances, including in human, can cause serious disorders or even arrest of embryo development, which determines the great practical interest for MZT studies. The MS under review is devoted to the comparative analysis of MZT timing and regulatory mechanisms in early human development, with special emphasis on the comparison with other animals, primarily mice. The author discusses not only some differences between various MRT strategies, but also the current possibilities of using a mouse model to improve the efficiency of ART. The submitted MS, therefore, is of great interest to a wide range of specialists.

The MS is conceived and written clearly and precisely. No principal questions or comments arise after reading. Meanwhile, it would be helpful to add a schematic drawing or table that could summarize and highlight the similarities/differences between human and mouse MZT. Such a final summary would further enhance the value of the review for readers.

Response: I have added two summary figures highlighting the differences between human and mouse MZT. Thank you for your comment.

Reviewer 2 Report

This manuscript presents an interesting topic of study. However, the title of manuscript is not consistent with the conclusions. In this sense, it should noted that the conclusions are exclusively focused on the mouse while the title is open to other animal species. Therefore, a deeper review should have been done in other species.

Author Response

Comments and Suggestions for Authors

This manuscript presents an interesting topic of study. However, the title of manuscript is not consistent with the conclusions. In this sense, it should noted that the conclusions are exclusively focused on the mouse while the title is open to other animal species. Therefore, a deeper review should have been done in other species.

Responses:

Actually, this paper was initially invited as a minireview. Later on, The Editoral Office  changed their proposition to a full review, adding 600 words to the main  text. I hope it’s OK now. Anyway, I have added, in the title (highlighted), that the manuscript is based on the comparison betwen human and mouse, the most available animal study model of those close to the human species. This does not exclude references to data obtained in other species, but they are mentioned only marginally.

Round 2

Reviewer 2 Report

I consider that new version of manuscript can be published.